# Breast cancer screening practices among Vietnamese women and factors associated with clinical breast examination uptake

Tran Thu Ngan[1,2]*, Chris Jenkins[1], Hoang Van Minh[2], Michael Donnelly[1], Ciaran O'Neill[1]

**1** Centre for Public Health, Queen's University Belfast, Belfast, United Kingdom, **2** Centre for Population Health Sciences, Hanoi University of Public Health, Hanoi, Vietnam

* n.t.tran@qub.ac.uk

**Data Availability Statement:** All relevant data are within the paper and its Supporting information files.

## Abstract

### Background

This study examined current breast cancer (BC) screening practices among Vietnamese women and the factors associated with the uptake of clinical breast examination (CBE).

### Methods

A total of 508 women aged 30–74 years in Hanoi completed a knowledge-attitude-practice (KAP) survey in 2019 including validated measures of breast cancer awareness (Breast-CAM) and health beliefs (Champion's Health Belief Model Scale). Descriptive statistics, $\chi 2$, and ANOVA tests were used to analyse KAP responses across groups with different socio-demographic characteristics. A logistic regression model assessed the associations of knowledge, beliefs, and sociodemographic characteristics with CBE uptake.

### Results

Only 18% of respondents were aware of BC signs, risk factors, and screening modalities although 63% had previously received BC screening. CBE was the most common screening modality with an uptake of 51%. A significantly higher proportion of urban residents compared with rural residents (32% vs 18%, Chi-square test, p = 0.04) received mammography. Unlike mammography, CBE uptake was not associated with sociodemographic characteristics (i.e., residence area/education level/occupation/household monthly income/possession of health insurance). CBE uptake was associated with BC knowledge (OR = 2.44, 95%CI: 1.37–4.32), perceived susceptibility to BC (OR = 1.15, 95%CI: 1.05–1.25), and perceived barriers to accessing CBE (OR = 0.88, 95%CI: 0.84–0.92).

### Conclusion

The study points to the need for public health education and promotion interventions to address low levels of awareness about BC and to increase uptake of BC screening in Vietnam in advance of screening programme planning and implementation. It also suggests that

**Funding:** The work reported in this paper was undertaken during TTN's PhD studies which was funded by the Profs Murray-Yarnell PhD studentship from the Faculty of Medicine and Health Sciences, Queen's University Belfast, United Kingdom (https://www.qub.ac.uk/). The funder provided support in the form of salaries for authors TTN, CJ, MD, and CON, but did not have any additional role in the study design, data collection and analysis, decision to publish, or preparation of the manuscript. The specific roles of these authors are articulated in the 'author contributions' section".

**Competing interests:** The authors have declared that no competing interests exist.

screening programmes using CBE are promising given current engagement and the absence of socio-demographic disparities.

## Introduction

Breast cancer (BC) is the most common cancer among Vietnamese women with 15,299 new cases in 2018 which accounted for 20.6% of all cancer cases in women [1]. This proportion is more than double that of the second most common cancer, colorectal (7,126 new cases, 9.6%) [1]. The estimated number of deaths due to BC was also the highest, at 6,103 deaths, which accounted for 13.9% of all cancer deaths [1]. Importantly, 64.7% of new BC cases were in women below the age 50 and 64.2% cases were diagnosed at late stage (stage III or IV) [2, 3]. Breast cancer in young women (aged < 40 years) tends to be more aggressive and diagnosed at later stages resulting in poorer survival rates compared to BC among older women [4, 5]. Although early detection through screening is critical in this context, Vietnam currently does not have a national BC screening programme.

Before implementing a screening programme, consideration needs to be given to the screening modality used, the feasibility of service delivery, the cost-effectiveness of the programme, and its acceptability to the targeted population. Mammography is very costly and prohibitively expensive in low-resource settings such as low-and middle-income countries (LMICs) [6, 7]. In contrast, clinical breast examination (CBE)–an alternative low-cost screening tool with downstaging effect–represents a realistic intervention in LMICs [8] and it has been shown to be cost-effective in Vietnam [9]. How acceptable such a programme would be is unclear though. This aspect is important to successful programme implementation. For example, a randomised controlled trial of CBE in the Philippines was terminated after the first screening round due to a low acceptance rate [10].

The Knowledge-Attitude-Practice (KAP) survey is a widely used method to improve understanding about health programme priorities and identify barriers to programme implementation [11, 12]. According to the World Health Organization (WHO), a KAP survey can identify knowledge gaps, beliefs, and behaviour patterns as well as the factors influencing these issues [11]. In KAP theory, the acquisition of knowledge, the generation of attitudes, and the formation of behaviours are three successive processes [13]. The KAP approach may draw upon relevant theories such as the health belief model (HBM) which points to the important influence of beliefs and perceptions in health harming and health promoting behaviours [14]. Therefore, the KAP approach and HMB were utilised in this study to assess the current screening practices of Vietnamese women and the factors associated with their uptake of CBE.

## Methods

### Study design, setting, and participants

We conducted a cross-sectional KAP household face-to-face interview survey at the community level in August 2019, in Hanoi which is the capital and the second most populous city in Vietnam.

The sample comprised (i) women aged 30–74 years who (ii) never had BC (self-reported), and (iii) consented to participate in the survey. Although BC screening programmes in Europe, the Americas, and Australia target women aged 40–74 years [15–18], this study chose the age range 30–74 years to reflect the younger age at which Vietnamese women, on average, are diagnosed with BC. In Vietnam, 64.7% of new BC cases in 2012 were below the age of 50

years old [2] and the age standardized incidence rate (ASR) increased from the age of 30 years old (i.e., 3.2, 20.7, and 54.5/100,000 women among the age group of 20–29, 30–39, and 40–49 years, respectively) [1].

This study received ethical approval (reference no: 319/2019/YTCC-HD3, dated 30 May 2019) from the Hanoi University of Public Health's Institutional Review Board. All respondents received an information sheet about the study including information about its voluntary nature, objectives, target respondents, privacy, use of collected data, potential drawbacks, and benefits of participation. Interviewers also provided respondents with a verbal explanation about the study's purpose and the interview procedure. Written informed consent was obtained from all respondents who agreed to participate.

## Sample size and sampling methods

Sample size was calculated according to WHO guidance and using a formula that estimated a population proportion with specific absolute precision [19] (S1 Table in S1 File). The estimated proportion of women who had had BC screening in Vietnam was assumed to be 50% to generate the most conservative, or largest, sample size. A sample size of 500 was calculated using a 95% confidence interval, an absolute precision value of 0.1, a design effect of 2 (to account for cluster sampling) and a non-response rate of 10%.

Multistage sampling was used to sample survey respondents. In stage 1, one urban and one rural district in Hanoi were randomly selected; in stage 2, two communes per district were selected using Population Proportionate to Size (PPS) sampling; in stage 3, every nth household (from a chosen starting point) was approached (interval = 2); and in stage 4, one eligible respondent per household was interviewed. Age quota (S2 Table in S1 File) was applied (calculated based on the female age structure of Vietnam from Census 2009 [20]). If the eligible respondent was not at home or was not able to participate at the point of interview, interviewers would re-visit one time.

## Survey tool

The Breast Module of the Cancer Awareness Measure (Breast-CAM, which can be accessed and downloaded freely at https://www.cancerresearchuk.org/health-professional/awareness-and-prevention/the-cancer-awareness-measures-cam) developed by Cancer Research UK, King's College London, and University College London in 2009 [21] was used to assess respondents' knowledge of BC. The Champion Health Belief Model Scale (CHBMS) was used to assess respondents' beliefs about BC/BC screening [14].

Breast-CAM was used to assess women's knowledge of the UK Breast Screening Programme. Currently, there is no BC screening programme in Vietnam. Therefore, Breast-CAM was modified to enquire about a respondent's knowledge of BC screening modalities. The CHBMS does not have a CBE-specific version (only a breast self-examination-BSE (version 1993) and a mammography (version 1999)) [14, 22]. Unlike BSE, mammography and CBE are screening modalities that need to be provided by clinicians. Thus, we replaced the word 'mammography' with 'clinical breast examination' in all items except one item of the CHBMS version 1999. This modification changed the focus of the items to CBE and did not alter the meaning or purpose of the items. The item, 'Having a mammogram exposes me to unnecessary radiation' was excluded. Cronbach's α for the CHBMS ranged from 0.75 to 0.88 in the original study [14] and from 0.69 to 0.78 in this study.

The instruments were translated into Vietnamese by the first author who is bilingual. Back translation was carried on by another researcher from Hanoi University of Public Health (HUPH). We compared and discussed the two versions and made some minor amendments.

This revised Vietnamese version of the questionnaire was piloted by interviewing 10 volunteer respondents in Hanoi. The results of the pilot interviews were used to finalise the questionnaire used in the main data collection exercise.

## Data collection

10 interviewers were recruited from HUPH's 3rd and 4th year student cohorts who had experience of working as community survey interviewers. Two research assistants at HUPH acted as field supervisors and assisted the interviewers. The team received a 1-day-training course prior to data collection. The training included an introduction to the study objectives, how to select households and respondents, questionnaire briefing, and a practice session.

Data collection was carried out on weekends (17-18/8/2019 for the rural district and 23-26/8/2019 for the urban district). In each district, two supervised teams of five interviewers conducted the interviews simultaneously in two selected communes of the district. Data collection lasted longer in the urban area as the teams could interview women only early in the morning or late afternoon instead of throughout the whole day as was possible in the rural areas (because the proportion of urban dwelling women working as full-time employees was higher).

## Variables and measurements

**Main outcome: CBE screening practice/uptake.** CBE screening uptake was a binary (yes or no) variable. A definition of CBE was provided alongside this question about CBE uptake to ensure that respondents had a common understanding of CBE and did not confuse it with BSE.

**Main predictors: Knowledge of BC and attitude/belief.** Questions were posed about three categories of knowledge: 'BC symptoms', 'BC risk factors', and 'BC screening modalities'. Women who had knowledge in all three components were defined as 'having knowledge' or being knowledgeable about BC. Regarding each component, women who identified more than five non-lump symptoms (out of nine) or risk factors (out of 10) were defined as 'having knowledge' of BC symptoms and BC risk factors, respectively [21]; women who named at least one correct screening modality without any prompting (e.g., mammography, CBE, breast ultrasound) were defined as 'having knowledge' of BC screening modalities.

Attitudes/beliefs were assessed by using the modified CHBMS version 1999; 18 items were grouped into three subscales, perceived susceptibility (3 items), perceived benefits (5 items), and perceived barriers (10 items). Survey participants chose one of five responses to each item: Strongly disagree (1)–Disagree (2)–Not sure (3)–Agree (4)–Strongly agree (5) and the score (from one to five) to each item was summed to calculate a total score for each subscale.

**Covariates.** The selection of covariates was based on the results of a systematic review that identified factors associated with the uptake of BC screening in China [23]: 'age', 'education level' (completed at least primary education/completed secondary education/completed high school education/completed university degree), 'occupation' (full-time employee/self-employed/housewife/retired), 'residence area' (urban/rural), 'possession of health insurance (HI)', and 'household's monthly income' (in six categories based on income quintile of general Vietnamese population in 2016 [24]).

## Data analysis

Descriptive statistics (mean, standard deviation-SD, min/max values for continuous variables and percentages for discrete variables) were used to describe the sociodemographic characteristics of respondents, their knowledge of BC, their attitudes/beliefs towards BC/BC screening,

**Table 1. Sociodemographic characteristics by area (urban vs rural).**

|  | Total | Urban | Rural | p-value |
|---|---|---|---|---|
|  | n (%) | n (%) | n (%) |  |
| **Total** | 508 (100.0) | 256 (50.4) | 252 (49.6) |  |
| **Age, mean (sd)** | 46 (11) | 47 (11) | 46 (11) | NS |
| **Education level** |  |  |  |  |
| Completed at least primary education | 107 (21.1) | 22 (8.6) | 85 (33.7) | <0.001 |
| Completed secondary education | 173 (34.1) | 61 (23.9) | 112 (44.4) |  |
| Completed high school education | 110 (21.7) | 81 (31.8) | 29 (11.5) |  |
| Completed university degree | 117 (23.1) | 91 (35.7) | 26 (10.3) |  |
| **Occupation** |  |  |  |  |
| Full-time employee | 98 (19.3) | 65 (25.4) | 33 (13.1) | <0.001 |
| Self-employed | 303 (59.6) | 122 (47.7) | 181 (71.8) |  |
| Homemaker/housewife | 66 (13.0) | 33 (12.9) | 33 (13.1) |  |
| Retired | 41 (8.1) | 36 (14.1) | 5 (2.0) |  |
| **Marital status** |  |  |  |  |
| Single/Separated/Divorced/Widow | 43 (8.5) | 19 (7.4) | 24 (9.5) | NS |
| Married | 465 (91.5) | 237 (92.6) | 228 (90.5) |  |
| **Household monthly income** |  |  |  |  |
| < = 3,000,000 VND (~$129[a]) | 34 (6.7) | 2 (0.8) | 32 (12.7) | <0.001 |
| 3,000,001–6,000,000 VND (~$130–259) | 72 (14.2) | 27 (10.6) | 45 (17.9) |  |
| 6,000,001–9,000,000 VND (~$260–389) | 70 (13.8) | 34 (13.3) | 36 (14.3) |  |
| 9,000,001–12,000,000 VND (~$390–519) | 140 (27.6) | 61 (23.9) | 79 (31.3) |  |
| 12,000,001–25,000,000 (~$519–1079) | 129 (25.4) | 85 (33.3) | 44 (17.5) |  |
| >25,000,000 VND (~$1079) | 42 (8.3) | 36 (14.1) | 6 (2.4) |  |
| **Possessed health insurance** | 396 (78.0) | 212 (82.8) | 184 (73.0) | 0.008 |
| **Ethnicity: Kinh** | 505 (99.4) | 255 (99.6) | 250 (99.2) | NS |
| **Religion: No religion** | 489 (96.3) | 242 (94.5) | 247 (98.0) | NS |

[a] Currency exchange rate in October 2020: 1 USD = 23,176 VND

NS: Not significant | VND: Vietnamese Dong (the currency of Vietnam) | $: United State Dollar (USD)

and their screening practice or use. Knowledge and practice across groups with different socio-demographic characteristics were assessed using the Chi-square test while ANOVA assessed between-group differences regarding attitudes/beliefs. The factors that influenced CBE uptake were investigated using a logistic regression model. The various statistical procedures were conducted in STATA version 15.0.

## Results

A total of 508 women completed the interviews (response rate of 95%). Only 21 out of 533 women refused to participate and four did not complete the interview. Respondents' sociode-mographic characteristics by area (urban vs rural) are presented in Table 1. Most respondents were married (92%); from the majority Kinh ethnic group (99%); and had no religion (96%). The average age of respondents was 46; 55% completed at least secondary education; 60% were self-employed; 71% had a household monthly income higher than 9,000,000 Vietnamese Dong (VND) (~$389); and 78% had HI.

## Knowledge of breast cancer

61% of respondents were knowledgeable about BC symptoms (i.e., they identified $\geq 5$ non-lump symptoms). The top three commonly reported symptoms were 'lump in breast' (85%), 'discharge from nipple' (79%), and 'pain in breast/armpit' (77%). The three least commonly known symptoms were 'puckering/dimpling of breast skin' (42%), 'nipple rash' (42%), and 'redness of breast skin' (43%) (S1 Fig in S1 File). Only 40% of respondents had knowledge of BC risk factors. The most commonly known risk factors were 'past history of BC' (83%) and 'having a close relative with BC' (59%). The least commonly known risk factors were 'having late menopause' (21%) and 'starting periods early' (17%) (S2 Fig in S1 File). Half of respondents (49%) had knowledge of BC screening modalities. The most commonly known screening modality was CBE (63%), followed by breast ultrasound (52%), and mammography (23%) (S3 Fig in S1 File).

Only 18% of respondents had knowledge of all three domains (symptoms, risk factors, and screening modalities) (Table 2). Overall, a higher level of knowledge about BC was associated

**Table 2. Knowledge of breast cancer by sociodemographic characteristics.**

| Characteristics (n = 508) | Overall knowledge of BC[a] | | p-value |
|---|---|---|---|
| | n | % | |
| **Total** | 91 | 17.9 | |
| **Residence area** | | | |
| Urban | 62 | 24.2 | <0.001 |
| Rural | 29 | 11.5 | |
| **Education level** | | | |
| Completed at least primary education | 13 | 12.1 | 0.002 |
| Completed secondary education | 21 | 12.1 | |
| Completed high school education | 26 | 23.6 | |
| Completed university degree and above | 31 | 26.5 | |
| **Occupation** | | | |
| Full-time employee | 25 | 25.5 | <0.001 |
| Self-employed | 41 | 13.5 | |
| Homemaker/housewife | 10 | 15.2 | |
| Retired | 15 | 36.6 | |
| **Household monthly income** | | | |
| < = 3,000,000 VND (~$129[b]) | 6 | 17.6 | 0.004 |
| 3,000,001–6,000,000 VND (~$130–259) | 7 | 9.7 | |
| 6,000,001–9,000,000 VND (~$260–389) | 7 | 10.0 | |
| 9,000,001–12,000,000 VND (~$390–519) | 25 | 17.9 | |
| 12,000,001–25,000,000 (~$519–1079) | 33 | 25.6 | |
| >25,000,000 VND (~$1079) | 13 | 31.0 | |
| **Age group** | | | |
| 30–39 | 30 | 17.2 | NS |
| 40–49 | 26 | 16.8 | |
| 50–59 | 16 | 15.0 | |
| 60–74 | 19 | 26.4 | |

[a] Knowledge of BC: Have knowledge in all of the following three domains 'symptoms', 'risk factors', and 'screening modalities'

[b] Currency exchange rate in October 2020: 1 USD = 23,176 VND

BC: Breast cancer | NS: Not significant | VND: Vietnamese Dong | $: United State Dollar (USD)

**Table 3. Mean CHBMS subscale scores by sociodemographic characteristics.**

| | Perceived susceptibility[a] mean (sd) | P-value | Perceived benefits[b] mean (sd) | p-value | Perceived barriers[c] mean (sd) | p-value |
|---|---|---|---|---|---|---|
| **Total** | 9.3 (2.3) | | 19.7 (2.1) | | 23.7 (4.8) | |
| **Residence area** | | | | | | |
| Urban | 9.2 (2.2) | NS | 19.6 (2.1) | NS | 22.5 (4.3) | <0.001 |
| Rural | 9.4 (2.4) | | 19.7 (2.1) | | 24.9 (5.0) | |
| **Occupation** | | | | | | |
| Full-time employee | 9.7 (1.9) | NS | 19.2 (2.2) | NS | 22.0 (4.3) | <0.001 |
| Self-employed | 9.4 (2.4) | | 19.7 (2.1) | | 24.3 (4.9) | |
| Homemaker/housewife | 8.9 (2.4) | | 20.0 (1.8) | | 24.2 (3.9) | |
| Retired | 8.8 (2.2) | | 20.2 (2.0) | | 22.6 (5.5) | |
| **Education level** | | | | | | |
| Completed at least primary education | 9.4 (2.7) | NS | 19.6 (2.2) | <0.001 | 26.2 (5.2) | <0.001 |
| Completed secondary education | 9.2 (2.3) | | 20.0 (1.9) | | 24.1 (4.7) | |
| Completed high school education | 9.4 (2.2) | | 20.0 (1.8) | | 22.6 (4.2) | |
| Completed university degree and above | 9.4 (2.1) | | 19.0 (2.3) | | 21.8 (4.1) | |
| **Household monthly income** | | | | | | |
| < = 3,000,000 VND (~$129[d]) | 8.8 (2.7) | NS | 19.6 (2.5) | NS | 27.6 (4.6) | <0.001 |
| 3,000,001–6,000,000 VND (~$130–259) | 9.3 (2.4) | | 19.8 (1.9) | | 25.1 (5.5) | |
| 6,000,001–9,000,000 VND (~$260–389) | 9.5 (2.2) | | 19.5 (2.0) | | 24.3 (4.5) | |
| 9,000,001–12,000,000 VND (~$390–519) | 9.4 (2.3) | | 20.0 (1.7) | | 23.9 (4.4) | |
| 12,000,001–25,000,000 (~$519–1079) | 9.6 (2.2) | | 19.5 (2.3) | | 22.4 (4.2) | |
| >25,000,000 VND (~$1079) | 9.3 (2.1) | | 19.3 (2.2) | | 21.3 (4.2) | |
| **Age group (in years)** | | | | | | |
| <40 | 9.7 (2.0) | <0.001 | 19.2 (2.3) | <0.001 | 22.8 (4.5) | NS |
| 40–49 | 9.5 (2.4) | | 19.8 (1.8) | | 24.3 (4.9) | |
| 50–59 | 9.0 (2.3) | | 19.8 (2.1) | | 24.5 (4.7) | |
| 60+ | 8.5 (2.5) | | 20.3 (1.8) | | 23.4 (5.4) | |

[a] Perceived susceptibility scale has min = 3, max = 15

[b] Perceived benefits scale has min = 5, max = 25

[c] Perceived barriers scale has min = 10, max = 50

[d] Currency exchange rate in October 2020: 1 USD = 23,176 VND

CHBMS: Champion Health Belief Model Scale | NS: Not significant | sd: standard deviation | VND: Vietnamese Dong (the currency of Vietnam) | $: United State Dollar (USD)

with living in an urban area, a higher education level, retirement status, and a higher household monthly income (Chi-square tests, all tests p<0.05). There was no significant association between knowledge of BC and age.

## Attitude/belief towards BC/BC screening

Table 3 shows subscale scores for the CHBMS by sociodemographic characteristics. Younger respondents had a significantly higher perceived susceptibility score regarding BC (ANOVA test, p<0.001) whereas older respondents had a significantly higher perceived benefits score for CBE (ANOVA test, p<0.001). Respondents who lived in a rural area, were self-employed (including homemaker/housewife), had a lower education level, and had a lower household monthly income were more likely to have significantly higher scores regarding perceived

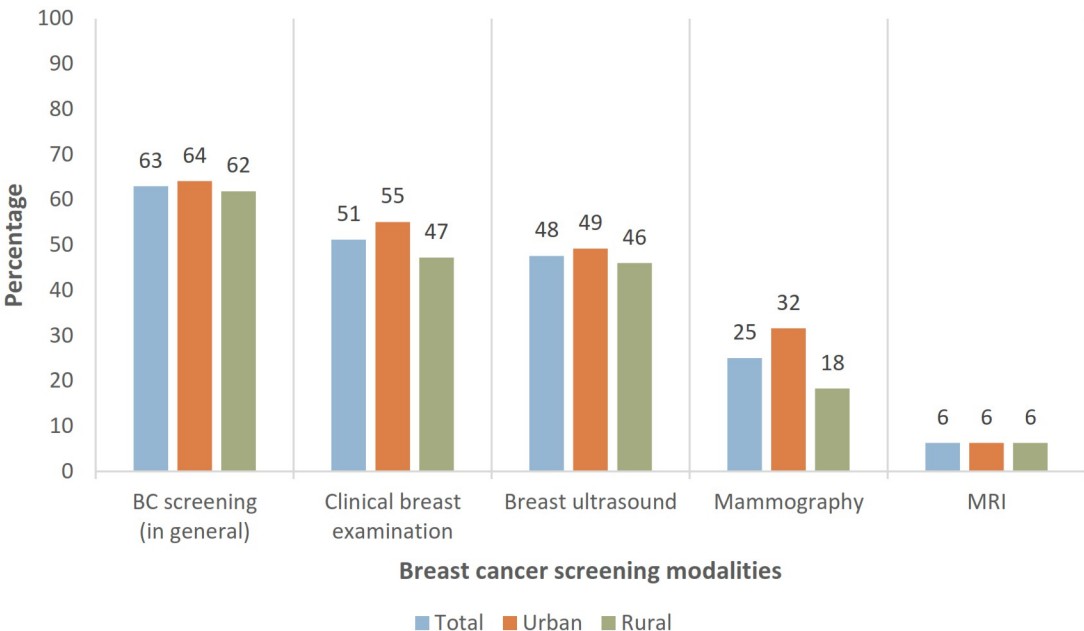

**Fig 1. Respondents who had been screened for breast cancer (any modality) by area.**

barriers to accessing and using CBE (ANOVA test, all tests p<0.001). Age was not associated with the perceived barriers subscale score.

## CBE screening practice/uptake

63% of respondents had experienced some mode of BC screening (Fig 1); 51% of this proportion reported that they had received CBE, followed by breast ultrasound (48%), mammography (25%) and MRI (6%). Area of residence was not associated with uptake except in the case of mammography—32% of respondents in urban area compared to 18% of rural dwellers had had a mammogram (Chi-square test, p = 0.04).

## Factors associated with CBE uptake

Table 4 shows the results of a logistic regression model that assessed the association between a range of factors and CBE uptake. Sociodemographic characteristics (i.e., residence area, education level, occupation, household monthly income, and possession of HI) and a respondent's perceived benefits of getting CBE were not associated with the uptake of CBE. Women who had knowledge of BC were 2.4 times more likely to avail of CBE (OR = 2.44, 95% CI: 1.37–4.32). Each point higher in the perceived susceptibility score significantly increased the odds of CBE uptake 1.15 times (OR = 1.15, 95% CI: 1.05–1.25) whilst each point higher in the perceived barriers score significantly decreased the odds of CBE uptake 0.88 times (OR = 0.88, 95% CI: 0.84–0.92).

## Discussion

The results of the analysis indicate a low level of overall BC knowledge (18%). Yet, 63% of respondents reported participating in at least one type of BC screening modality. Among all types of BC screening modalities, CBE had the highest uptake—two times higher than

**Table 4. Which factors are associated with the uptake of clinical breast examination?**

| Characteristics (n = 504) | Ever had CBE Mean (sd) | Adjusted Odds ratio[a] | 95% CI |
|---|---|---|---|
| **Perceived susceptibility of getting BC** (min = 3, max = 15) | **9.7 (2.2)** | **1.15**[*] | **[1.05–1.25]** |
| **Perceived benefits of getting CBE** (min = 5, max = 25) | 19.8 (2.0) | 1.03 | [0.94–1.13] |
| **Perceived barriers of getting CBE** (min = 10, max = 50) | **22.5 (4.5)** | **0.88**[**] | **[0.84–0.92]** |
| | **Ever had CBE n (%)** | **Adjusted Odds ratio[a]** | **95% CI** |
| **Overall knowledge of BC** | | | |
| No[ref] | 195 (46.8) | 1.00 | |
| Yes | **65 (71.4)** | **2.44**[*] | **[1.37–4.32]** |
| **Residence area** | | | |
| Urban[ref] | 141 (55.1) | 1.00 | |
| Rural | 119 (47.2) | 1.22 | [0.78–1.92] |
| **Age groups** | | | |
| <40[ref] | 96 (55.2) | 1.00 | [1.00–1.00] |
| 40–49 | 79 (51.0) | 1.14 | [0.69–1.89] |
| 50–59 | 58 (54.2) | 1.36 | [0.74–2.47] |
| 60+ | 27 (37.5) | 0.47 | [0.21–1.01] |
| **Education level** | | | |
| Completed at least primary education[ref] | 37 (34.6) | 1.00 | |
| Completed secondary education | 88 (50.9) | 1.61 | [0.90–2.88] |
| Completed high school education | 62 (56.4) | 1.71 | [0.84–3.49] |
| Completed university degree | 72 (61.5) | 2.01 | [0.88–4.60] |
| **Occupation** | | | |
| Full-time employee[ref] | 61 (62.2) | 1.00 | |
| Self-employed | 145 (47.9) | 0.92 | [0.50–1.70] |
| Homemaker/housewife | 29 (43.9) | 0.83 | [0.37–1.82] |
| Retired | 25 (61.0) | 1.70 | [0.65–4.43] |
| **Household monthly income** | | | |
| < = 3,000,000 VND (~$129[b])[ref] | 10 (29.4) | 1.00 | |
| 3,000,001–6,000,000 VND (~$130–259) | 35 (48.6) | 1.41 | [0.54–3.69] |
| 6,000,001–9,000,000 VND (~$260–389) | 32 (45.7) | 1.06 | [0.39–2.82] |
| 9,000,001–12,000,000 VND (~$390–519) | 81 (57.9) | 1.49 | [0.60–3.65] |
| 12,000,001–25,000,000 (~$519–1079) | 71 (55.0) | 0.90 | [0.35–2.35] |
| >25,000,000 VND (~$1079) | 24 (57.1) | 0.86 | [0.28–2.63] |
| **Possession of health insurance** | | | |
| No[ref] | 46 (41.1) | 1.00 | |
| Yes | 214 (54.0) | 1.12 | [0.68–1.82] |

BC: Breast cancer | CBE: Clinical breast examination | CI: confidence interval | sd: standard deviation | VND: Vietnamese Dong (the currency of Vietnam) | $: United State Dollar (USD)

[ref]: reference group

[a] Odds ratios were adjusted for knowledge of BC, perceived susceptibility, perceived benefits, perceived barriers, residence area, age group, education level, occupation, household monthly income, and possession of health insurance.

[b] Currency exchange rate in October 2020: 1 USD = 23,176 VND

[*] $p < 0.05$,

[**] $p < 0.001$

mammography. CBE uptake was not associated with sociodemographic characteristics in contrast to mammography. This finding together with the results of other research [8] indicates the potential for CBE to be used to extend access to essential cancer services in an equitable way.

Regarding CBE uptake, this study reported a higher percentage compared with previous studies among women in general aged 20–60 years conducted in 2015 and 2017 (51% vs 32%) [25, 26]. The higher CBE uptake proportion in our study may be related to the influence of previous pilot CBE screening programmes in Hanoi (which was the study site for 4/5 pilots [27]) or may reflect a trend towards increased knowledge. Studies in other specific Vietnamese populations such as ethnic minorities and female teachers in primary schools [28, 29] have reported wider variation in CBE uptake, from 17% to 63%. Regarding mammography uptake, both current and previous studies reported a significantly lower uptake proportion compared to CBE [25, 29, 30].

The study also highlights the strong association between BC knowledge and uptake of CBE. Although previous studies of the association between knowledge and CBE specifically are limited, the link between knowledge and BC screening uptake in general is well established [23, 28, 31, 32]. Our results regarding the other factors that were associated with CBE uptake i.e., perceived susceptibility and perceived barriers, highlight the importance of ensuring that a BC screening programme is targeted towards populations that have received BC awareness raising and, so, are informed about and have a good understanding of BC in order to help to maximise the potential of the screening programme.

The study underscores that the current level of BC knowledge (across all three categories: 'symptoms', 'risk factors', and 'screening modalities' among Vietnamese women aged 30–74 years in Hanoi is extremely low (18%). The use of self-developed questionnaires and various definitions of 'good/bad knowledge' in all previous KAP studies in relation to BC in Vietnam (five in English and 12 in Vietnamese, published between 2009 and 2019) makes it difficult to undertake meaningful comparisons between the results of these studies and the study that is presented in this paper and, in addition, hinders an assessment of cancer awareness and knowledge in Vietnam including trends over time. Future research should use standardised instruments in order to generate reliable, comparative, and actionable findings to inform public health and cancer preventive service planning decisions.

Several socio-demographic factors were associated with knowledge of BC. Women who lived in an urban area, who had a higher education level and higher household monthly income, and who were retired were more likely to have better knowledge about BC. This pattern of results is consistent with similar studies, globally [23, 25, 33–35] and point to the need for BC awareness raising programmes that are targeted towards, and tailored to, particular groups in the population of Vietnamese women.

Regarding factors that were associated with beliefs towards BC/CBE, the influence of age (group) on the sub-scales of the CHBMS is notable. Indeed, the age of a woman had a negative association with her perceived susceptibility to BC, but it had a positive association with the benefits that she perceived came from CBE screening, whilst age did not appear to exert any influence on her perception of the barriers to availing of CBE. Arguably, this particular mixture or combination of beliefs is likely to lead women to engage in screening behaviour, particularly if they participate in BC knowledge and awareness raising programmes [36]. The much younger age at which BC is diagnosed in Vietnam compared to HICs [2, 7, 37] may explain why younger respondents were more alert to the likelihood of a BC diagnosis despite the fact that BC risk increases with age [38]. The finding that socio-demographic factors such as living in a rural area, being self-employed/a housewife, having lower education and household

income were significantly associated with the perception that there were barriers to CBE screening points again to the need for a targeted public health/cancer education programme.

The study has several strengths. For example, it is the first study in Vietnam to use standardised and validated instruments (i.e., Breast-CAM and CHBMS) to assess BC knowledge, attitudes, and beliefs among Vietnamese women. As such, it facilitates comparisons with studies from other countries and provides reliable data for the planning of BC interventions and related policy in Vietnam. More specifically, the study provides novel and valuable insights about the factors that influence the uptake of CBE which will contribute to the implementation of future screening programmes. The study's limitations include uncertainty about the extent to which the results may be generalized to the whole country and, so, further research is required. We could not explore the influence of ethnicity in relation to CBE uptake as 99% of respondents were from the majority Kinh ethnic group. There is a need for further research to investigate KAP and uptake among ethnic minorities in Vietnam (there are 53 ethnic minorities and they account for 15% of the population [39]).

## Conclusions

Only 18% of Vietnamese women aged 30–74 years old had knowledge of BC symptoms, risk factors, and screening modalities though around 63% had had previous experience of BC screening. CBE was the most common screening modality (51% of screened women). Mammography tends to be located in the larger medical centres and it is unsurprising, perhaps, that the uptake in urban areas was almost double the proportion that was reported by respondents in rural areas. Unlike mammography, CBE uptake was not associated with sociodemographic characteristics. CBE uptake predictors were knowledge of BC, perceived susceptibility to BC, and perceived barriers to using CBE. Public health education or promotion interventions are essential preceding the implementation of a BC screening programme in Vietnam. Current engagement and the absence of socio-demographic disparities indicate that a CBE programme is likely to produce positive outcomes for Vietnamese women, their families, and wider society.

## Supporting information

**S1 Appendix. Minimal dataset.**
(XLS)

**S1 File. Supporting tables and figures (containing S1, S2 Tables and S1-S3 Figs).**
(PDF)

## Acknowledgments

The authors express our sincerest thanks to the Centre for Population Health Sciences (Hanoi University of Public Health), Hanoi Department of Health, Cau Giay District Health Center, and Quoc Oai District Health Center for their support in the data collection process.

## Author Contributions

**Conceptualization:** Tran Thu Ngan, Michael Donnelly, Ciaran O'Neill.

**Formal analysis:** Tran Thu Ngan.

**Funding acquisition:** Michael Donnelly, Ciaran O'Neill.

**Investigation:** Tran Thu Ngan.

**Methodology:** Tran Thu Ngan.

**Resources:** Hoang Van Minh.

**Supervision:** Hoang Van Minh, Michael Donnelly, Ciaran O'Neill.

**Writing – original draft:** Tran Thu Ngan.

**Writing – review & editing:** Tran Thu Ngan, Chris Jenkins, Hoang Van Minh, Michael Donnelly, Ciaran O'Neill.

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
