## [Decision Letter · Decision Letter 0]

22 Feb 2022

PONE-D-21-17006

Breast cancer screening practices among Vietnamese women and factors associated with clinical breast examination uptake

PLOS ONE

Dear Dr. Tran,

Thank you for submitting your manuscript to PLOS ONE. After careful consideration, we feel that it has merit but does not fully meet PLOS ONE’s publication criteria as it currently stands. Therefore, we invite you to submit a revised version of the manuscript that addresses the points raised during the review process.

The manuscript has been evaluated by two reviewers, and their comments are available below.

The reviewers have raised a number of concerns that need attention. They request additional information on methodological aspects of the study (such as participant recruitment and the scale used). They also request improvements to the discussion of the results in the context of the existing literature.

Could you please revise the manuscript to carefully address the concerns raised?

We look forward to receiving your revised manuscript.

Kind regards,

Marianne Clemence

Associate Editor

PLOS ONE

Journal Requirements:

2. Thank you for including your ethics statement: "The study received the ethical approval No.319/2019/YTCC-HD3 dated 30th May 2019 from Hanoi University of Public Health’s Institutional Review Board. Informed consent was obtained from all individual participants included in the study." 

a) Please provide additional details regarding participant consent. In the ethics statement in the Methods and online submission information, please ensure that you have specified what type you obtained (for instance, written or verbal, and if verbal, how it was documented and witnessed). If your study included minors, state whether you obtained consent from parents or guardians. If the need for consent was waived by the ethics committee, please include this information.

3. Thank you for stating the following financial disclosure: "The work reported in this paper was undertaken during TTN’s PhD studies which is funded by the Profs Murray-Yarnell PhD studentship from Faculty of Medicine and Health Sciences, Queen’s University Belfast, United Kingdom (https://www.qub.ac.uk/). The funder had no role in study design, data collection and analysis, decision to publish, or preparation of the manuscript."

We note that one or more of the authors is affiliated with the funding organization, indicating the funder may have had some role in the design, data collection, analysis or preparation of your manuscript for publication; in other words, the funder played an indirect role through the participation of the co-authors. If the funding organization did not play a role in the study design, data collection and analysis, decision to publish, or preparation of the manuscript and only provided financial support in the form of authors' salaries and/or research materials, please do the following:

a. Review your statements relating to the author contributions, and ensure you have specifically and accurately indicated the role(s) that these authors had in your study. These amendments should be made in the online form.

b. Confirm in your cover letter that you agree with the following statement, and we will change the online submission form on your behalf: 

“The funder provided support in the form of salaries for authors [insert relevant initials], but did not have any additional role in the study design, data collection and analysis, decision to publish, or preparation of the manuscript. The specific roles of these authors are articulated in the ‘author contributions’ section.

5. Please note that in order to use the direct billing option the corresponding author must be affiliated with the chosen institute. Please either amend your manuscript to change the affiliation or corresponding author, or email us at plosone@plos.org with a request to remove this option.

Reviewers' comments:

Reviewer's Responses to Questions

**Comments to the Author**

1. Is the manuscript technically sound, and do the data support the conclusions?

Reviewer #1: Yes

Reviewer #2: Partly

2. Has the statistical analysis been performed appropriately and rigorously? 

Reviewer #1: Yes

Reviewer #2: Yes

3. Have the authors made all data underlying the findings in their manuscript fully available?

Reviewer #1: Yes

Reviewer #2: No

4. Is the manuscript presented in an intelligible fashion and written in standard English?

Reviewer #1: Yes

Reviewer #2: Yes

5. Review Comments to the Author

Reviewer #1: The subject matter of this manuscript is a global public health issue and very critical for all countries. The authors have presented the results in an intelligible fashion in mostly simple and clear language. The sample size is large and seem representative for the Vitanamese women of reproductive age. The results are well analyzed and presented clearly and the statistical analysis are robust. The discussion section is also presented concisely and the conclusions are drawn based on the results presented.

I do have a few suggestions that I think will make the manuscript even more meaningful to all readers.

1. Please, provide more elaboration on the ethical review and approval process in addition to the approval number and the statement that participants gave consent prior to be being included in the study. What kind of consent was given? Written or verbal and how was participants informed and convinced to participate and to consent?

2. Please, provide some explanation on the inclusion and exclusion criteria for the study besides the age limitation.

3. In the results presentation, you used statements such as " approximately, less than, about around two-thirds". Please, be straight forward and use the exact percentages instead of the probable phrases before the percentages in parenthesis.

4. Please, be consistent with the use of either Vietnam women or Vietnamese women instead of using them interchangeably. However, I could be wrong as both could be acceptable in Vietnam. Please, advise.

5. There are a few vocabulary issues and a careful edit will resolve them, please.

6. In the discussion section, please, provide more relevant studies in the literature related to your study and do the relevant comparisons.

7. Please, add a section on strengths and limitations and add the statement on your use of standardized instrument compared to personally developed instruments as a strength. Then provide a weakness if there a challenge you encountered.

Overall, the study is important and will provide a critical resource for health promotion and disease prevention officers in addition to adding knowledge to existing literature.

Reviewer #2: Given the absence of a copy editor, it would serve the authors well to have the manuscript proof-read and edited to improve conciseness.

Abstract:

Was the instrument cross culturally adapted, translated, back translated? Please elaborate.

Was IRB approval obtained, were patients consented. Based on answers to questions, it appears they did. But might be better to state this in methods: In this IRB-approved study, consented patients responded to a 300-item scale cross culturally adapted from the …”

When you say, “18% of V. Women had knowledge’” what does that mean – this percentage discussed with someone, heard about breast cancer, …

Introduction:

Excellent first paragraph with high yield information. It might help to mention that in the absence of a population-based screening program, early detection (of which awareness is key) is essential.

Second paragraph:

The second sentence is inaccurate and should be removed. In fact, mammography is highly useful in this age group. It does have reduced sensitivity in dense breast women, which is more common in Vietnamese and young women for a variety of reasons. However, it is still effective. Also, there are multiple reasons to list to describe why mammography is not a good option in LMICs.

Unlike first paragraph, where authors provided specific data to support their research, the second paragraph lacks data and speaks generally. I would prefer more information here. I think a lot of this paragraph is unnecessary and could be more concise to present data without increasing word count.

Methods:

Paragraph 1:

Please state why you chose neighborhoods in Hanoi and what you did when someone wasn't home (did you return, leave your number). What days and times of days did you visit? (if you visited during weekday, then maybe women at home were more likely new mothers or unemployed, or…). It helps reader understand bias that might be present/how generalizable the results are.

I think the ethical statement should be at front of paragraph, but this is more stylistic.

Was the instrument translated, back translated, who translated? Were the interviewers trained, what experience did they have?

Results:

It would help to see the instrument. I’m not sure what a “non-lump symptom” is or what the authors considered risk factors.

Pain is a late symptom. In fact, studies have shown that pain alone as a symptom is unlikely to be related to breast cancer. Is this what you want women, particularly young women, to learn. Part of raising awareness is to limit overburdening the healthcare system and teaching “pain” as a breast cancer symptom could have opposite effect.

Breast cancer knowledge is presented as a scale. Did they check the relationship between the items?

Discussion:

Please use first paragraph to state the most important finding of the study and what implication this has on the Vietnamese population (as it pertains to breast cancer)

Authors do a good job of stating their results and stating (usually) how this relates to Vietnamese population, but then fail to drive home the “so what”. Also, the discussion lacks conciseness and synthesis of the results. I would avoid showing data here unless it helps convey a point you make in the paragraph.

6. PLOS authors have the option to publish the peer review history of their article (what does this mean?). If published, this will include your full peer review and any attached files.

Reviewer #1: No

Reviewer #2: No

---

## [Author Response · Author response to Decision Letter 0]

1 Apr 2022

Please find details in attached file 'Response to reviewers'

---

## [Decision Letter · Decision Letter 1]

22 Apr 2022

PONE-D-21-17006R1Breast cancer screening practices among Vietnamese women and factors associated with clinical breast examination uptakePLOS ONE

Dear Dr. Ngan,

Thank you for submitting your manuscript to PLOS ONE. After careful consideration, we feel that it has merit but does not fully meet PLOS ONE’s publication criteria as it currently stands. Therefore, we invite you to submit a revised version of the manuscript that addresses the points raised during the review process. Please submit your revised manuscript by June 06, 2022. If you will need more time than this to complete your revisions, please reply to this message or contact the journal office at plosone@plos.org. Please include the following items when submitting your revised manuscript:A rebuttal letter that responds to each point raised by the academic editor and reviewer(s). You should upload this letter as a separate file labeled 'Response to Reviewers'.A marked-up copy of your manuscript that highlights changes made to the original version. You should upload this as a separate file labeled 'Revised Manuscript with Track Changes'.An unmarked version of your revised paper without tracked changes. You should upload this as a separate file labeled 'Manuscript'.If applicable, we recommend that you deposit your laboratory protocols in protocols.io to enhance the reproducibility of your results. Protocols.io assigns your protocol its own identifier (DOI) so that it can be cited independently in the future. For instructions see: https://journals.plos.org/plosone/s/submission-guidelines#loc-laboratory-protocols. Additionally, PLOS ONE offers an option for publishing peer-reviewed Lab Protocol articles, which describe protocols hosted on protocols.io. Read more information on sharing protocols at https://plos.org/protocols?utm_medium=editorial-email&utm_source=authorletters&utm_campaign=protocols.

We look forward to receiving your revised manuscript.

Kind regards,

Muhammad Shahzad Aslam, Ph.D.,M.Phil., Pharm-D

Academic Editor

PLOS ONE

Journal Requirements:

Reviewers' comments:

Reviewer's Responses to Questions

**Comments to the Author**

1. If the authors have adequately addressed your comments raised in a previous round of review and you feel that this manuscript is now acceptable for publication, you may indicate that here to bypass the “Comments to the Author” section, enter your conflict of interest statement in the “Confidential to Editor” section, and submit your "Accept" recommendation.

Reviewer #1: All comments have been addressed

Reviewer #2: All comments have been addressed

2. Is the manuscript technically sound, and do the data support the conclusions?

Reviewer #1: Yes

Reviewer #2: Yes

3. Has the statistical analysis been performed appropriately and rigorously? 

Reviewer #1: Yes

Reviewer #2: Yes

4. Have the authors made all data underlying the findings in their manuscript fully available?

Reviewer #1: Yes

Reviewer #2: Yes

5. Is the manuscript presented in an intelligible fashion and written in standard English?

Reviewer #1: Yes

Reviewer #2: Yes

6. Review Comments to the Author

Reviewer #1: This is a much-improved version and a very sound and important research. In order to improve the manuscript even better and increased the possibility of its acceptance, I suggest the following.

1. Please, all your tables need to be appropriately formatted. Look up examples of APA table format and use it. It is much better presented than your current tables.

2. You need to provide subheadings for all the variables or analysis in the findings section. These include knowledge of BC, Health Model, CBE screening, and logistic regression model results.

3. In the findings section line 225, you indicated F1 but there no figure and there is no description. Please, delete or clarify

4. Using i.e should be avoided. Instead, write it in full

5. You need to be consistent with spelling regimes. Use programme throughout or program but do not in a single manuscript.

6. On line 290, use the term differentiated public health education programme. A more appropriate terminology is targeted public health where populations are divided into segments depending on some of criteria.

Overall, it is a very necessary and time research.

Reviewer #2: The authors wrote a substantially improved version of their prior manuscript. Small stylistic suggestion would include writing in the active voice. For example, rather than writing, "Instead, clinical breast examination (CBE) – an alternative low-cost screening tool with downstaging effect – is more likely to represent a realistic intervention in LMICs [8] and it has been shown to be cost-effective in Vietnam [9].", I suggest writing, "In contrast, clinical breast Examination (CBE) represents...".

Second sentence of discussion: Consider specifying which types of screening (All types/modalities?) you include in this percentage. For example, "Among all types of breast cancer screening,..." or "More than 63% of participants reported participating..."

7. PLOS authors have the option to publish the peer review history of their article (what does this mean?). If published, this will include your full peer review and any attached files.

Reviewer #1: No

Reviewer #2: No

---

## [Author Response · Author response to Decision Letter 1]

26 Apr 2022

Please find details in attached file 'Response to reviewers'.

---

## [Decision Letter · Decision Letter 2]

18 May 2022

Breast cancer screening practices among Vietnamese women and factors associated with clinical breast examination uptake

PONE-D-21-17006R2

Dear,

We’re pleased to inform you that your manuscript has been judged scientifically suitable for publication and will be formally accepted for publication once it meets all outstanding technical requirements.

Kind regards,

Muhammad Shahzad Aslam, Ph.D.,M.Phil., Pharm-D

Academic Editor

PLOS ONE

Additional Editor Comments (optional):

Reviewers' comments:

Reviewer's Responses to Questions

**Comments to the Author**

1. If the authors have adequately addressed your comments raised in a previous round of review and you feel that this manuscript is now acceptable for publication, you may indicate that here to bypass the “Comments to the Author” section, enter your conflict of interest statement in the “Confidential to Editor” section, and submit your "Accept" recommendation.

Reviewer #1: All comments have been addressed

Reviewer #2: All comments have been addressed

2. Is the manuscript technically sound, and do the data support the conclusions?

Reviewer #1: Yes

Reviewer #2: Yes

3. Has the statistical analysis been performed appropriately and rigorously? 

Reviewer #1: Yes

Reviewer #2: Yes

4. Have the authors made all data underlying the findings in their manuscript fully available?

Reviewer #1: Yes

Reviewer #2: Yes

5. Is the manuscript presented in an intelligible fashion and written in standard English?

Reviewer #1: Yes

Reviewer #2: Yes

6. Review Comments to the Author

Reviewer #1: I have thoroughly reviewed the manuscript again comparing this version to the original and first revised versions and have concluded that the authors have done a great job of improving the manuscript greatly. All the concerns I had and the almost all the minor issues have been addressed.

Reviewer #2: The authors addressed all my comments and the paper reads substantially better. They now provide substantial details necessary to replicate their work. Their work is important and will be interesting to the PLOS readership. There remain a couple of typos (For example):

1. In methods, first section line 87 should say "on average" because all Vietnamese women aren't diagnosed with breast cancer at an earlier age that European women, it's just that the average age at presentation is younger.

2. Line 92 "This study" rather than "The study"

3. first paragraph of discussion, last sentence should specify what "this screening" means - is it CBE or mammography. It may mean that a couple of commas are needed around in contrast to mammography. Also, do the authors mean "socially acceptable method for screening" (Rather than potential screening method) since their data shows that many women of all demographic received CBE despite few having awareness?

7. PLOS authors have the option to publish the peer review history of their article (what does this mean?). If published, this will include your full peer review and any attached files.

Reviewer #1: No

Reviewer #2: No

---

## [Editor Report · Acceptance letter]

20 May 2022

PONE-D-21-17006R2 

Breast cancer screening practices among Vietnamese women and factors associated with clinical breast examination uptake 

Dear Dr. Ngan:

I'm pleased to inform you that your manuscript has been deemed suitable for publication in PLOS ONE. Congratulations! Your manuscript is now with our production department. 

Kind regards, 

on behalf of

Dr. Muhammad Shahzad Aslam 

Academic Editor

PLOS ONE